# Faster, Higher, More Moral: Human Enhancement and Christianity

**Michael Buttrey** [1,*], **Moira McQueen** [2] **and Tracy J. Trothen** [3]

1 Regis College, University of Toronto, Toronto, ON M5S 2Z7, Canada
2 University of St. Michael's College, University of Toronto, Toronto, ON M5S 1J4, Canada; moira.mcqueen@utoronto.ca
3 School of Religion and School of Rehabilitation Therapy, Queen's University, Kingston, ON K7L 3N6, Canada; trothent@queensu.ca
* Correspondence: michael.buttrey@mail.utoronto.ca

**Abstract:** The three authors of this article explore the intersection of moral enhancement, ethics, and Christianity. Trothen reviews the meaning and potential of moral enhancements, considering some of the risks and limitations. Trothen identifies three broad ethical questions, which all three authors agree upon, that arise from a Christian theological perspective: what it means to be human, choice, and social justice. Trothen concludes that respect for human dignity and social justice requires rejecting a reductive view of moral improvement as purely biochemical. Buttrey then argues that biomedical moral enhancement (BME) is simply one in a series of attempts to morally improve human beings and can be compared to other efforts such as neo-Aristotelian virtue ethics. He argues that BME cannot be simultaneously more reliable than moral education in virtue and no more restrictive of human freedom. He concludes by suggesting that tensions between BME and Thomistic virtue are even stronger due to Christian conceptions of martyrdom and radical self-denial. Finally, McQueen argues that Christianity emphasizes the common good and social justice as essential for human flourishing. Building on the foundation established by Trothen and Buttrey, McQueen insists that accurate cognitive knowledge is needed to make good conscience decisions, but emphasizes that right human action also requires the exercise of the will, which can be undermined by AI, automation, and perhaps also BME. She concludes by encouraging further attention to the true nature of human agency, human freedom, and wisdom in debates over AI and biomedical enhancement. The authors conclude that BMEs, if they become medically safe, may be theologically justifiable and helpful as a supplement to moral improvement.

**Keywords:** moral bioenhancement; Christianity; ethics; virtue; dignity; justice; empathy

## 1. Introduction

As enhancement technologies increase, we will be able to make ourselves stronger, calmer, able to think more quickly and sharply, maybe even have more intense and transformative spiritual experiences, and possibly become more virtuous with the help of biomedical moral enhancements (BME). Artificial Intelligence (AI) will take care of many everyday tasks. We are developing computer programs and data banks that provide much more thorough and complex problem solving and give us ways to quickly process complex questions and problems. In the face of increasing human enhancement technologies (HET), we need to become more intentional about what it is that really matters to us—what we *really* value and desire. This intentionality involves deliberate self-reflexive work concerning our values and the effects of our social locations and experiences in shaping our values and worldviews.

In this article, we consider biomedical moral enhancement (BME) through a Christian theological lens. Our goal is to introduce BME, and some of the important ethics issues associated with BME arising at the intersection of moral enhancement and Christianity.

We do not claim to provide an exhaustive discussion, but we hope to introduce some significant challenges to the possibility of intentional moral bioenhancement and stimulate further conversation. This is not merely a thought experiment. As will become clear, many potential BMEs are already available and are being used for therapeutic purposes other than moral bioenhancement.

In her section, Trothen reviews the potential of biomedical enhancement (BME) and some of its risks and limitations. She explores the contextuality of morality, including the contextual meanings of empathy and assertiveness. Trothen argues that the Christian doctrine of the imago Dei affirms human dignity and implies an appreciation for interdependence. Providing a basis for Buttrey's exploration of virtue and freedom, she emphasizes the need for self-reflexivity in order to fully understand the context of human freedom and contends that collaborative design is necessary for biomedical enhancement to serve social justice, instead of magnifying prejudices. Finally, Trothen concludes that respect for human dignity requires rejecting a reductive view of moral improvement as purely biochemical.

In his section, Buttrey demonstrates how many BME proponents acknowledge standing in a long tradition of attempts to morally improve human beings, both to validate BME and to demonstrate the need for it. He argues that BME cannot be simultaneously more reliable than moral education and no more restrictive of human freedom, but also that analogies to moral education and biomedical therapies justify some forms of BME. Buttrey then summarizes two traditional views of virtue, Aristotelian and Thomistic, before comparing them to BME. Drawing on neo-Aristotelians, including Julia Annas, he argues that virtue is an intelligent skill, and therefore, it is at a higher level and more flexible than BMEs. Next, Buttrey argues that virtue's intelligence, flexibility, and relative rarity means it cannot be dismissed by simple observational studies. He is more concerned about virtue's tendencies towards elitism and ableism. Buttrey argues the Thomistic virtue tradition can remedy these problems by providing a balanced account of divine grace and human freedom. He concludes by suggesting that the tensions between BME and Thomistic virtue are even stronger due to Christian conceptions of martyrdom and radical self-denial. Buttrey's arguments add depth to the overarching argument in this article regarding the relevance of virtue and social justice to the intersection of Christianity and BME.

In her section, McQueen first questions fear-based arguments for BME. Next, she builds on the preceding discussions, arguing that Christian theology and church structure should lead to an emphasis on social justice and the common good as central concepts in human flourishing. McQueen then reviews invasive and non-invasive methods of brain stimulation, and their medical and potentially moral uses. She also examines Kevin Warwick's work on brain to computer interfaces, and other proposals to integrate technology with the human brain. McQueen then reflects theologically on whether we can expect computer algorithms to improve moral judgements. She argues that accurate objective facts are needed by the conscience to make good decisions, suggesting that cognitive enhancements that simply improve access to information would be helpful. However, she emphasizes that right human action also requires the exercise of the will, which can be undermined by AI, automation, and perhaps BME as well. She concludes by encouraging further attention to the true nature of human agency, human freedom, and human wisdom in debates over artificial intelligence and biomedical enhancement.

The three authors of this article served on the Faith and Life Science Reference Group of the Canadian Council of Churches at the time of the writing of this article. These three ethicists are informed in part by their diverse Christian traditions, respectively: The United Church of Canada, The Anglican Church, and The Roman Catholic Church. All three authors are committed to an ecumenical approach that respects diverse Christian traditions, as well as other religious traditions. Their diverse Christian identities and inclusive commitments have assisted them in introducing a range of Christian theo-ethical issues related to BME. The main objective of this article is modest: to serve as a starting point to further research and discussion.

## 2. Introducing Moral Bioenhancement and Christianity, by Tracy J. Trothen

### 2.1. Why Moral Bioenhancements?

Proponents of BME hope that these biomedical technologies will better the world by improving moral reasoning (and possibly altruism (Van Eyghen 2021)) through cognitive enhancement, increasing prosocial behaviors, strengthening motivation to do good, and/or enhancing moral virtues (Harris 2016; Douglas 2008; Hughes 2017). Well-known philosophers, Ingmar Persson and Julian Savulescu, advocate for the use of BME as a safeguard against the destructive potential of fast developing technologies that could be used to obliterate the planet (Persson and Savulescu 2008, 2012). They reason that with the proliferation of technologies, and especially cognitive enhancements, we will have more opportunities and tools for inflicting mass destruction. Consider, for example, autonomous weapons such as flying drones with sniper sensor devices and facial recognition software that tells these drones whose skulls to penetrate, and how to evade bullets. A professor of computer science at Berkeley warns that these unstoppable "slaughterbots" are not merely a science fiction-based figment of our imagination: slaughterbots could be created now by integrating tech that we already have in miniature form (Dust 2019).

Might Persson and Savulescu, and others including bioethicist James Hughes who writes from a Buddhist perspective (Hughes 2013), be right to say that biomedical means have the potential to help us become more moral in how we use technology? Particularly in the context of growing secularism in parts of Europe and North America, BMEs may become important augmentations to moral education. Historically, the church and other faith groups have served as focal points for many communities, providing moral guidance and social cohesion. Many places that have experienced a decline of organized religion may have yet to fill the moral guidance hole left behind. In some cases, social media has become a strong source of inconsistent and questionable moral guidance. An era of "fake news" has added to this instability. Can BMEs help to strengthen altruism, social justice, and lessen hostility, or are we looking for an easy way out of a problem that has no quick fix? BMEs may be consistent with some Christian perspectives, *if* moral education is used alongside these bioenhancements, BMEs are tailored to individuals and their social contexts, and BMEs are medically safe.

Persson and Savulescu may well be correct in arguing that our collective sense of justice, and inclination to care for each other, needs improvement to minimize the possibility of mass destruction. There is much room for global moral improvement in the context of war, human rights atrocities, abuse, and systemic ecological harm including climate change. The question is how best to do this from an informed Christian perspective.

### 2.2. How Might We Bioenhance Our Morality?

Since morality is partly neurobiological, morality can be affected by pharmacological interventions. For example, the stimulant drug methylphenidate, sold under trade names including Ritalin and Concerta, reduces impulsive aggression (Douglas 2008, p. 233). Methylphenidate may also contribute to moral enhancement by sharpening focus and augmenting problem-solving abilities, possibly including one's capacity to do ethical analyses. Modafinil, sold under the trade name Provigil, and the hormone oxytocin, increase empathy, cooperation, trust, and concentration. Serotonin may increase empathy and aversion to harming others.

Potential non-pharmacological MBEs include deep brain stimulation (DBS), transcranial magnetic stimulation (TMS), and transcranial direct current stimulation (tDCS), which are all types of brain stimulation that may increase cooperation (Piore 2015) and neuroplasticity. Greater neuroplasticity could enhance one's ability to learn prosocial behaviors such as cognitive empathy and decrease impulsive aggression. Later in this article, McQueen further explores the possible role of brain stimulation in moral enhancement.

Genetic modification technologies may be the next frontier for moral and emotional improvement. Military organizations have expressed interest in neuroscience involving the alteration of genes that influence traumatic memory formation (Tennison and Moreno 2012;

Pitman et al. 2002). Traumatic memories can deter people from repeating or even initially engaging in violence. This is only a subset of the technologies that are emerging as possible ways to improve morality.

### 2.3. Limitations and Risks of BMEs

There are limitations and risks associated with the potential use these medical treatments as BMEs, instead of for their primary therapeutic purposes. Oxytocin increases empathy but only towards ingroup members, or kin. It is possible that an increased empathy to kin could increase hostility toward those who are perceived as outgroup members (Trothen 2017a, 2017b). Persson and Savulescu suggest that this limitation of oxytocin could be mitigated by moral reasoning and education: "we contend that this restrictive tendency can be counteracted by moral reasoning and, thus, that BME 'would have to go hand in hand with reasoning which undercuts race, sex, etc. as grounds for moral differentiation'" (Persson and Savulescu 2019, p. 816). Although moral education may suffice to make some MBEs justifiable and even helpful, ingroup/outgroup thinking seems to have a strong instinctual and emotional rootedness that may not be sufficiently overcome by an education in moral reasoning to the degree needed to make oxytocin more helpful than harmful. The tendency to distrust outgroup members has been well established as part of a human survival instinct. This distrust may lead to hostility if this "natural disposition" to distrust is not addressed constructively (Van Eyghen 2021, p. 8). Although cognitive enhancements may help address this distrust, few people may be willing to do the necessary educational and self-reflexive work to overcome such biased thinking.

Potential BMEs also have health risks in spite of their therapeutic value. As McQueen discusses later in this article, DBS is a successful treatment for many people with Parkinson's Disease, intractable clinical depression, and other conditions, but may cause seizures or headaches, and affect personal identity in unforeseen ways by possibly changing thought patterns (Cabrera et al. 2014). Although these biomedical technologies successfully treat some serious medical conditions, there is no consensus regarding the potential efficacy of these technologies for the primary purpose of improving morality.

### 2.4. The Contextual Nature of Morality

Morality is very difficult to define. Ought the focus be on moral behaviours, on motives that underlie behaviours (Douglas 2008), on virtues such as altruism and justice (Persson and Savulescu 2012; Hughes 2013), or on the improvement of cognitive abilities that could improve ethical reasoning (Harris 2011)? One reason morality is difficult to define is its contextuality. As Buttrey probes in the next section of this article, the meaning of doing good (Hauskeller 2016) and of virtues such as altruism or justice changes a bit or a lot depending on the situation and who is involved (Jones 2013, p. 150; De Melo-Martin and Salles 2015; Hauskeller 2016).

For example, empathy and self-sacrifice may not be virtues for every person in every situation. Empathy is not "uncontroversial" as has been claimed by some in the BME conversation (Persson and Savulescu 2012, p. 409). Feminist scholars have established that marginalized people often are socialized to be overly altruistic and self-sacrificing to the neglect of adequate self-love. In addition, psychologists have demonstrated that socio-economically privileged people are in greater need of enhanced empathy compared to those with less money (Piff et al. 2012; Kraus et al. 2011). Although pride may be the prominent sin for many privileged people, the prominent sin for many marginalized people is often the loss of self, power, and voice (Saiving 1960). Assertiveness, self-pride, self-interest, and even some aggression may be what needs to be enhanced to make some people more virtuous, if dignity and social justice are valued.

### 2.5. Non-Biomedical Tech Moral Improvement? Robotics, Empathy Labs, and Spirituality

Perhaps a safer and less controversial alternative to moral enhancement may be through narrow AI. Robots increasingly are being used in healthcare to provide caring

support. "Robot pet therapy" helped during the COVID pandemic when many hospital and long-term care patients had little outside contact with loved ones (Knibbs 2020). Other robots, such as Pepper, have been around longer. As Pepper explains, "I was created in 2014 by Softbank Robotics in Tokyo, Japan. I flew all the way to Toronto to work at Humber River Hospital—I love it here". Equipped with sensors and cameras, Pepper is designed to detect emotions and respond supportively. Robots can model caring and compassionate behavior, encouraging recipients to do likewise. Robots may be able to model a degree of empathy, but robots are limited. Human touch and compassion cannot be entirely replaced (Purtill 2019). Spiritual distress, including existential angst, will not be addressed fully by robots but could be alleviated in a supplemental capacity.

Empathy labs, designed based on neurological and other social-scientific research, are another emerging moral enhancement modality (Trothen 2016). Moreover, spirituality, too, is associated with increased compassion and prosocial behavior toward strangers (Saslow et al. 2013). Traditional and new ways of enhancing spirituality may also serve to enhance morality (Mercer and Trothen 2021, pp. 115–40).

*2.6. Values and Social Processes: Technology Is Not Value Neutral*

How do we decide which, if any, technologies should be used as a means of moral enhancement? Much of this conversation comes down to our values and the belief systems and worldviews that shape our values. Values "pertain to beliefs and attitudes [and ideals] that provide direction to everyday living" (Corey et al. 2014, p. 14). Values are complicated. It is tough to name and unpack the social processes that influence what we value and desire (Sherwin 2012). What do we really, really want? European philosophers Marcuse (1964), Habermas (1971), and Foucault (1988) have made a strong case that technology (which includes moral bioenhancements as applied science) is not value neutral, but that technology promotes the values of utility and efficiency. Not only are we bombarded by advertising messages telling us what we desire, technology may contribute to the often false belief that utility and efficiency are more important than other values such as relationships, social justice, and environmental health. (A quick way to help unearth some of the things that are most important to us is to imagine what you want in your obituary; how do you want to be remembered?).

*2.7. Introducing BME and Christianity*

What can religion bring to this conversation? For followers of religions, values are strongly influenced by the central stories and doctrines of the religions. There are many points of intersection between Christianity and BME. Three significant theological intersections are: interdependence as a key aspect of what it means to be human, the meaning of choice, and social justice. My goal is to identify some of these intersections and begin to suggest theological issues relevant to the creation and use of BMEs. My co-authors deepen the meaning of these issues.

2.7.1. Theological Anthropology: BMEs and What It Means to Be Human

In Christianity, the doctrine of the imago Dei affirms the inherent dignity and value of humanity. What it means to be human—and the best humans we can be—is at the heart of the moral enhancement conversation. Debate regarding what it means to be made in the image of God partly reflects differing views on the nature of God. Although some have placed an emphasis on a capacity for rational thought, others disagree, seeing the essence of God—and humans as created in God's image—as more about relationality (Trothen 2017a, 2017b). The latter view has gained support in the last half century, including from relational, liberation, and feminist theologians. A relational interpretation of the imago Dei doctrine emphasizes covenant and interdependence. Christian theologian Philip Hefner proposed the interpretation that people are created co-creators (Hefner 1993), with a moral and theological duty to work for the improvement of life. Although Hefner's theological anthropology has been critiqued as being too trusting of human freedom

and the human capacity to do good, his proposal aligns well with more liberal Christian theologies that emphasize potential human goodness. The "created" aspect of Hefner's proposal acknowledges the Christian belief in human capacity to create harmfully, either intentionally or accidentally.

A relational interpretation of the imago Dei includes a recognition of the interdependence of life. This recognition shifts the focus of the moral enhancement debate from the individual to the enhancement of communities. An emphasis on individual rights gives way to an emphasis on responsibility. Part of this responsibility requires self-reflexive awareness of systemic power and inequitable access to resources.

Theological anthropology embraced the steps that are considered to make humans better. Later in this article, McQueen develops the theological claim that humans are created with intrinsic dignity and responsibility for one another, in relation to BMEs. Buttrey adds to this conversation by asking what it means to be a morally improved human being; however, a thorough exploration of Christian theological anthropology is beyond the scope of this article. For our purposes, it is sufficient to introduce the question of what it means to be human as a question that arises at the intersection of BME and Christianity.

### 2.7.2. The Meaning of Choice

There is much debate about choice in discussions of BME. There is concern regarding the freedom to choose to be morally enhanced and the freedom to make authentic moral choices following enhancement. Will our individual choices reflect individual dignity and contribute to a socially just world? An elite athlete may not choose to be less aggressive but may choose to become better able to evaluate when aggression is needed. A very selfish person would likely not choose to become more altruistic unless it would benefit them. It may be that those who need BMEs most would not choose them. For those who would embrace BMEs, would their subsequent choices—after being morally enhanced—be in keeping with their authentic selves (Harris 2011), or would those subsequent choices merely reflect the influence of a drug or other BME? Maybe, by pharmacologically removing or minimizing neurological barriers, BMEs will be able to improve our capacity to make choices that are more authentic to who we truly want to be and were created to be (Hughes 2017).

Given the complex social processes influencing our values and desires, we need to ask to what degree are we truly free to choose (Sherwin 2012; Marcuse 1964; Habermas 1971; Foucault 1988). Self-reflexivity may be most effectively cultivated through education and emerging techniques such as empathy labs (Trothen 2016), than it may be through pharmacological or neurological means such as brain stimulation. Self-reflexivity requires awareness of the social, political, and cultural factors that help to form experiences. An awareness of the ways in which systemic dynamics shape power and affect perceptions of the self and others is necessary to becoming better morally, if moral is assumed to have a communal dimension. Without self-reflexivity, choices and relationships will suffer from the uncritical projection of what are assumed to be normative experiences. McQueen calls this capacity for self-critique and attentiveness to global implications, conscience, and examines the role of conscience further in her section. Although BMEs may not be able to replace traditional modes of moral development, including education about marginalization and equity, BMEs may assist moral capacity development in a supplementary manner. BMEs that increase attention or computational power, combined with traditional and innovative educational approaches, may help us to further self-reflexivity. Buttrey furthers this discussion regarding the complex relationship between education and BMEs in the following section.

### 2.7.3. Social Justice

If interdependence is understood as a virtue instead of an undesirable limitation, implications for community rather than the individual must be the starting point for ethical discussions about BME (Trothen 2017a, 2017b). Relationship is a central feature of

covenantal Christian theology, as is social justice and the Roman Catholic concept of the common good, as McQueen elucidates later in this article. Questions regarding equity and access to enhancing resources assume a high importance if human dignity is respected and individual well-being is understood to be firmly connected to communal well-being (we have only to consider the COVID-19 pandemic to see how inequitable access to vaccines affects global health).

Technology alone, no matter how accessible, will not get rid of bias. Algorithms such as those used in facial recognition software often reflect the biases of coders and may magnify social marginalization patterns. In their subsequent sections, Buttrey and McQueenexpand on social justice and offer further analysis of systemic bias as it relates to BME. Who decides what is "normal" and desirable and what needs to be "fixed" has a bearing on which virtues are assumed to be virtuous and which are seen as undesirable. Even when some virtues are agreed on in principle, such as dignity, compassion, and social justice, the meaning and application of these virtues varies, depending on context and community.

Co-design—meaning the inclusion of diverse voices at the creation stage of technology—is a principle consistent with the Christian commitment to the last being first. The epistemological prioritization of those at the social margins is core to Christian interpretations of social justice; the voices of those living on the margins are believed to be necessary to knowing what is wrong socially and understanding how to address these injustices.

### 2.8. Concluding Comment

Respect for human dignity and social justice suggest that a reductionist approach to humanity is to be rejected. Humans are more than enhance-able components and cranial neuro-pathways (Wiseman 2016). Attempts to make humans better morally will fall short if moral improvement is assumed to be a biochemical task only. BMEs cannot replace moral formation, disciplines, and other education regarding self-reflexivity; however, BMEs could be congruent with some liberal interpretations of Christianity if they are medically safe, socially just, and considered as supplementary to other forms of moral enhancement.

## 3. "Moral Improvement, Old and New" by Michael Buttrey

### 3.1. Introduction

Persson and Savulescu's first publication on biomedical moral enhancement (BME) call it "moral enhancement" (Persson and Savulescu 2008). Later works by them and other proponents add the prefixes "bio" or "biomedical", as in "biomedical moral enhancement" (BME) or "moral bioenhancement". Why was the modifier added? Due to the fact that there is a long history of attempts to morally improve human beings, a history that can be read to validate or vitiate proposals for BME; thus, distinguishing BME from these other attempts became necessary.

For my section, I will be comparing biomedical moral enhancement (BME) to two traditional approaches to moral improvement: Aristotelian virtue ethics and Thomistic virtue. I will start with how proponents of BME compare their proposals to traditional moral education, then present a contemporary account of Aristotelian virtue, and finish with a brief discussion of Christian virtue. My thesis is that biomedical moral enhancement proponents are right to suggest there are limits to our biological capacity for virtue and that virtue risks elitism; contemporary virtue theorists are right to insist on the potential of Aristotelian virtue for moral improvement; and the Christian virtue tradition reconciles these insights in an egalitarian, non-elitist fashion by balancing acquired and infused virtue.

### 3.2. Moral Enhancement vs. Moral Education

Many proponents of BME use moral education as a point of comparison for their proposals. According to Persson and Savulescu, "it is obvious that moral enhancement by traditional, cultural, means—i.e., the transmission of moral instruction and knowledge from earlier to subsequent generations—has not been as effective and quick as cognitive

enhancement by these means" (Persson and Savulescu 2008, p. 168). Thus, they suggest medical and genetic treatments may be faster and more accessible methods of BME. At the same time, they use the comparison to education to defend BME against the charge that it could restrict freedom. As they put it, "there is no reason to assume that moral bioenhancement to which children are exposed without their consent would restrict their freedom and responsibility more than a traditional moral education to which they are also exposed without their consent" (Persson and Savulescu 2008, p. 113). For Persson and Savulescu, the traditional moral education of children does not necessarily restrict their freedom, but it is also not effective at producing right action. In contrast, they claim that BME will be more productive of right action and yet no more restrictive than moral education.

Writing at the same time as Persson and Savulescu, Thomas Douglas also draws on an analogy to moral education to defend BME. Douglas is particularly interested in the possibility of reducing counter-moral emotions like racial aversion and impulsive violent aggression. He argues that even if such bad motives are 'natural' or part of an agent's given identity, "the appropriate attitude to take towards such properties is precisely one of *non-acceptance* and a *desire for self-change*" (Douglas 2008, p. 235). Douglas then contends that if such a change can be accomplished through moral education or BME, there is no reason to prefer the former or see the latter as more 'unnatural'. If human goal-directed action makes a change unnatural, then BME through self-improvement is just as artificial, and if it is rather the use of technology that is unnatural, then medical treatments for diseases are just as objectionable.

A later thinker, Mark Walker, proposes a 'Genetic Virtue Project'. He argues that vices and virtues are at least partially inherited, and if we can identify the genes responsible, we could use genetic selection and engineering to reduce vice and promote virtue. Walker's comparison to moral education arises in response to the objection that genetic moral enhancements would be much harder to change than traditional virtues. Walker's reply is that he sees no reason why new moral insights could not lead to new genetic moral enhancements; in addition, "children born with the wrong enhancements could be sent to remedial camps" (Walker 2009, p. 36). Walker does not reflect further on this chilling suggestion, and he also does not acknowledge differences between education and genetic engineering. One difference is that students who are educated in a certain way may individually choose to resist and reject the moral framework they were given, whereas individuals who were engineered to favour a certain morality will need the assistance of willing physicians, geneticists, and/or pharmaceutical companies to make further changes to their genetic inheritance.

Finally, Barbro Fröding sees the potential for moral enhancement to complement education in virtue. In her words, "to be vicious is to be irrational as such agents subscribe to mistaken beliefs about the good life" (Fröding 2011, p. 228); therefore, she believes cognitive enhancement in particular could be conducive to the development of virtue, by enabling less biased judgements, and encouraging a deeper understanding of virtue. Moreover, given that most people are subject to "substantial cognitive constraints", Fröding believes cognitive enhancement may be an essential tool to elevate them to the intellectual level where they have a good opportunity to develop the virtues needed for the good life; BME is therefore a remedy for the difficulties and elitism within traditional virtue education.

To some extent, some of these arguments are convincing. For example, Persson and Savulescu's claim that BME will be more effective than moral education, but no more restrictive of human freedom, is difficult to sustain. Human freedom is a major reason why traditional moral education is such an unreliable method of producing right action, and presumably it will be a major obstacle for effective BME as well. Failing to acknowledge this and a failure to explain how freedom can be protected without equally impeding the prospects of BME, is a major oversight by Persson and Savulescu. Similarly, although both education and genetic engineering of children requires the contributions of other

people, Walker does not consider how the latter will involve experts with more specialized knowledge, more sophisticated tools, and more financial and ideological commitments than the average parent or teacher. The required involvement of technical experts does not necessarily make genetic engineering more nefarious than traditional moral education, but it will make it harder for individuals or small groups to deviate from an elite moral consensus.

On the other hand, Douglas is right that the analogies to moral education and medical therapies justify some forms of BME. Brain surgery or genetic engineering is far more invasive than parents verbally correcting their children, but a hypothetical drug that could reduce episodes of impulsive violent aggression is not that different from treatments we generally accept, such as stimulants to treat ADHD or SSRIs for depression. Similarly, although accusing most human beings of cognitive constraints is a poor rhetorical strategy for overcoming elitism, I agree with Fröding that BME is better understood as a complement to virtue rather than a simple replacement for it.

Building on Trothen's discussion, next, I probe the meaning of "traditional moral education", and how it results (or not) in moral improvement. In the next two sections I summarize two understandings of virtue—neo-Aristotelian and Thomistic Christian—and consider how they compare to these BME proposals and if they address some of these concerns.

### 3.3. New Aristotelians, New Virtues

Neo-Aristotelian virtue theory is a contemporary reimagining of Aristotle by scholars such as Julia Annas (Annas 2011). According to Annas, developing virtues does not depend on the circumstances of your life but how you live it. Your circumstances include anything that is not under your control, such as age, gender, height, family, nationality, language, and so on. These circumstances influence your life and potential, but the measure of virtue is how skillfully (or not) you use your 'raw materials'. Moreover, until we all have do-it-yourself genetic engineering kits, genes will remain a greater part of our circumstances compared with how we live our lives. Thus, even if it helped provide a better starting point, Walker's genetic virtue project could not result in true virtue; developing virtue would still depend on what you did with your genetic gifts.

Like Aristotle, Annas argues that virtues are intelligent skills. That is, the brave person is guided by a rational understanding of bravery, traditionally understood as a context sensitive middle point between cowardice and recklessness. If so, then Thomas Douglas's proposal to medically suppress counter-moral emotions like racial aversion, even if potentially helpful, would still need to be complemented by an intelligent understanding of racial tolerance for true virtue.

Finally, Annas contends that virtues, like other skills, are initially learned by imitation, but the goal is for students to appropriate and understand the skill to the point that they may surpass their teacher. In contrast, pharmaceuticals and other medical technologies are usually designed for uniform effects. This difference identifies a significant difference between moral education and BME and reveals how the uniformity of BME may be more restrictive of freedom than virtue.

Even if virtue is clearly distinct from biomedical moral enhancement, does it work? Some critics claim that traditional moral education has been shown to be ineffective. For example, a study of thousands of American schoolchildren in the 1970s found a lack of consistent altruism, honesty, and self-control between situations (Hartshorne and May 1975). Similarly, a study at Princeton Seminary found that students who were late to give a talk were less willing to stop and help someone groaning in an alley, even if the subject of the talk was the parable of the Good Samaritan (Darley and Batson 1973). A third study often cited is the famous Milgram experiment, where subjects complied with instructions to administer simulated shocks up to a supposedly dangerous level (Milgram 1963). In light of these, some have argued that moral behaviour is primarily situational rather than reflective of any underlying character, and suggested Aristotelian virtues may not exist.

Virtue theorists interpret these studies differently (Annas 2005; Croom 2014). First, each of these studies found that some people acted well, regardless of situation. Second, Aristotle believed virtue is rarely found in children, because they lack life experience; this is also true of students, and so virtue being rare in these groups is to be expected. Third, if Annas is right, virtue involves the ability to vary actions intelligently in different situations, which makes it hard to measure empirically. Put another way, to accurately judge if someone is acting virtuously or viciously, you need to examine the reasons for their actions, not just their outward behaviour. For example, the seminary students may have passed by because they were trying to keep a promise to their professors and fellow students. The subjects in the Milgram study may have thought they could trust a biology teacher at Yale University to be honest and not let them hurt anyone. These reasons do not excuse their actions, but they suggest that simple observation is insufficient.

Another critique of virtue is Barbro Fröding's claim that virtue is elitist and too difficult. For Aristotle, who lived 24 centuries ago, one's potential for virtue depended on being born as an intelligent free man with good parents and a good society. However, Annas, writing in the 21st century, takes the more egalitarian route of emphasizing how you live your life over its circumstances. Virtue is also more egalitarian than BME because its price is not set by pharmaceutical companies or insurance providers, and its availability is not determined by market forces. Finally, even if BME were cheap or covered by universal healthcare, it would still be elitist in the sense that only some people in some countries would have the power, resources, and skills to choose the goals of BME, design biomedical moral enhancements, and decide who receives them. Every agent inherits an understanding of virtue from their parents, teachers, and society, but through life experience, self-reflection, and rational understanding, every agent has the freedom to adapt the virtues to their life and context as they develop them. Unless every human being is able to receive the education and tools to do their own genetic engineering or drug design, BME will inevitably be less accessible and less flexible than virtue.

Nevertheless, Annas agrees with Aristotle that children are incapable of virtue, because they lack the life experience needed for practical wisdom. Annas' account also implies that people with intellectual disabilities cannot be virtuous. Is there an alternative to excluding these two groups?

### 3.4. Thomistic Virtue and Conclusion

A full analysis of the Christian virtue tradition is beyond the scope of this paper, but I will highlight some initial insights. The Christian theologian with the greatest influence on the virtue tradition is Thomas Aquinas. In his *Summa Theologiae*, Aquinas agrees with Aristotle that the proper end of human life is happiness, but interprets true happiness as the beatific vision of God. Furthermore, he argues that moving towards this happiness requires gifts from God: the theological virtues of faith, hope, and charity (ST I-II 62.3). As these virtues are given, not earned, their acquisition does not depend on having certain capacities, so children and people with intellectual disabilities can be virtuous. Of course, any appeal to the ability of divine action to infuse virtue may seem like a form of special pleading that merely confirms the ableism of virtue, suggesting that although most people can acquire virtue themselves, people with disabilities need help. A better way to understand infused virtue begins with Alasdair MacIntyre's insight that "[t]here is no point in our development towards and in our exercise of independent practical reasoning at which we cease altogether to be dependent on particular others" (MacIntyre 1999, p. 97). In other words, dependency is not a rare state, found only among those with disabilities, but a fundamental aspect of the human condition. This observation of universal dependency is in tension with the Enlightenment model of the self-sufficient rational agent, but it harmonizes well with Aquinas' theological claim that all human beings depend on gifts from God to achieve human happiness; thus, a properly Thomistic theological anthropology sees all human beings as being in a position of *dis*ability to achieve true virtue without God, not just those people we see as having particular disabilities.

Nevertheless, understanding theological virtues as divine gifts naturally invites the question about whether Christians have human reasons to pursue moral improvement. Aquinas says yes: even with grace, fully restoring human nature requires the repeated cooperation of human freedom over time, to acquire an easy facility with virtue (ST I-II 65.3ad2). Famously, the Reformer Martin Luther was deeply suspicious of even the potential of practices such as reading scripture or prayer to help acquire virtue. He held the belief that sanctification, similarly to justification, requires total human passivity, so that God can work in us unhindered (Herdt 2008). Calvinism, with its stress on limited election and double predestination, undermines the motivation for good works even further (Herdt 2008); however, theologians such as Augustine, Jonathan Edwards, and N.T. Wright are more positive about moral virtue. For example, Augustine sees spiritual exercises, and industriously "checking and lessening . . . greed", for temporal things as good methods for moral and spiritual development—if they are driven by a passionate love for God (De Trin. 14.23); therefore, both Augustinian and Thomistic accounts of sanctification preserve motivations for moral improvement.

Finally, although BME may be used by Christians to supplement virtue development, the history of Christianity suggests that tensions between BME and virtue may be even stronger with Christian views of virtue. For example, many Christians venerate martyrs who were killed for their faith, and the early church tradition of ascetic monasticism persists until this day; however, it seems unlikely that many government agencies or pharmaceutical companies will fund research into biomedical moral enhancements that encourage martyrdom or monasticism. Similarly, we may ask if modern parents will see extreme generosity, radical self-denial, or even a passionate commitment to justice as good goals for BME or as distractions from achieving a successful life; therefore, although Christians may use BMEs to supplement the virtue development of themselves and their children, they will have to carefully discern whether a given BME has the flexibility to serve their understanding of the good life, or whether it risks further co-opting them into a model of goodness chosen and codified by the BME designers.

## 4. Technological Bio-Enhancement and Moral Agency, by Moira McQueen

### 4.1. Introduction

Persson and Savulescu's call for biomedical moral enhancement (BME) seems to be based on fear: fear that, in the face of the destruction of our planet as climate changes caused by humanity increase and multiply, we do not have the moral knowledge and will to enact changes because of our limited regard for others (Persson and Savulescu 2012). They believe more moral education is necessary, since not enough is being done at present, with changes advancing faster than humans' capacity to respond. If enhancement through external means could enhance our capacity for achieving results faster, then they believe they should be used. They feel the need to 'put these ideas on the table' because they view the situation as desperate.

Many questions can be asked here. Do Persson and Savulescu paint a realistic picture or are they being overly negative? As Trothen points out, climate change dangers are real, and dramatic changes have already taken place, e.g., receding glaciers, increase in extinction of some species, encroachments and destruction of rainforests and woodlands, land scarring from mining and oil production, etc. It is true that awareness of these factors is growing, but the will to combat them seems to be lacking, if, for example, lack of compliance with measures such as reduction of carbon emissions by every country can be taken as indicative. On the other hand, there have been many lifestyle changes globally, even if they have been inadequate so far, but could we be encouraged to be more intentional in our approach? There is a touch of 'Nostradamus' in Persson and Savulescu's approach, but fear of the threat of calamity is not always the best motivator. Savulescu often claims that we find it easier to harm people than to help them and this might appear to be accurate, given the scale of enmity, warfare, and colonization that has gone on since the beginning of

recorded history, all of which militate against people working together to save the planet and all that is within it.

### 4.2. Is There a Christian Alternative to Fear as Motivator of 'Better' Moral Thinking?

The story of humankind, however, also reveals other aspects of human nature which Christianity aims to foster. Along with our propensity towards enmity, there is a seemingly natural urge to be *better* persons in the moral sense; we *do* often recognize that our neighbour is as important as ourselves; there *is* a concept of the common good. These are acknowledged to be human, experiential and relational realities and many national, legal, and parliamentary systems are built on forms of social contracts which try to embody these realities. One of the theological mainstays of Catholic social teaching, for example, is that the pursuit of life and flourishing is not only about the individual but also about the common good. The two are tied: one aspect cannot flourish properly if the other aspect is ignored. Christianity itself is a global, communal organization centred on membership in the Body of Christ. If any people should be leading the way in emphasizing the need for the common good, it should be Christians! What are we, as Christians, saying about our responsibility for the common good in response to our common ecological problems? Much has been written by various denominations about these matters, (e.g., by Francis 2015), whereas Persson and Savulescu and others believe our response could be greatly improved through the use of biomedical moral enhancements. Those enhancements must be ethically evaluated to show how such moral improvement would be achieved (i.e., are the methods used ethical?) and whether they contribute to the moral growth of the individuals concerned and the common good (i.e., are they effective?).

### 4.3. Various Means

In our discussions, Trothen shows, among other points, the results of using pharmacology on moral thinking, and asks salient questions about the social justice aspects of BME, stating that 'the voices of those on the margins are necessary to know how to avoid injustice in these areas. This fits with my denomination's emphasis on 'the preferential option for the poor', which is hard to exercise in these discussions, especially in the world of 'high tech'which informs much of our paper. These concerns must be raised as a counter to questions about elitism and ableism in accessing possible treatments for cognitive and/or biomoral enhancement, noted by Buttrey.

Buttrey's comparison of BME with traditional methods of moral education emphasize the necessity of virtue in its many forms and how we can become more moral (i.e., morally enhanced), but not through the use of BME, which use external means. All our denominations take the Christian approach that knowledge without virtue is not enough: our actions must reflect Christian values, and achieving those values takes multiple virtues: discipline in learning and research; wisdom in reflecting and concluding; altruism in transcending subjective desires on consideration of the common good and social justice, and so on, within the context of following the Way of Jesus Christ in loving God and loving our neighbour as ourselves. That is a far cry from a more reductionist approach found in some 'research imperative' or moral compulsion models of achieving better moral thinking.

In my part of the paper, I would like to comment, first, on current technological means of improving cognition and ascertaining behaviour; and second, whether they might also be capable of enhancing moral thinking and decision-making.

### 4.4. Technological Means

Concerning the first part, there are several technological methods currently in use to improve cognition with application in healthcare to remedy the effects of stroke, dementia, and other neurological problems caused by illness or accident. Some of these have already been indicated by Trothen, and not all are currently fully approved by the FDA, since some are considered to be still in the experimental stage, although they have been shown to be relatively safe and effective.

Transcranial Magnetic Stimulation, Deep Transcranial Magnetic Stimulation, and Deep Brain Stimulation

The most commonly used method is transcranial magnetic stimulation (TMS), which is non-invasive and uses electromagnetic induction. An insulated coil is placed over the scalp on the part of the brain believed to be involved in mood regulation. Magnetic pulses about equal to those used in magnetic resonance imaging (MRI) are repeated rapidly (repetitive TMS or rTMS) and can make lasting changes to brain activity. This method was approved by the FDA in 2008 as a safe treatment for people with depression, and especially for those for whom antidepressants have severe side effects (Johns Hopkins Hospital n.d.).

The Johns Hopkins Brain Stimulation Program uses a coil "designed to affect extensive neuronal pathways, including deeper cortical regions and fibers targeting subcortical regions, without a significant increase in the electric field induced in superficial cortical layers." (Johns Hopkins Hospital n.d.).

Another method, deep brain stimulation (DBS), has shown improvement in clinical trials for mood and cognitive disorders such as major depression and Alzheimer's disease. Electrodes are implanted in specific regions targeting the underlying cause of the disease. Although studies have shown that DBS can reverse blood flow changes, much like antidepressant medication, it is not currently FDA approved for depression or bipolar disorder. It is approved for use in essential tremor, Parkinson's disease, dystonia, and chronic and severe obsessive-compulsive disorder, use for the latter being the first psychiatric indication to win approval.

Johns Hopkins Hospital also uses transcranial direct current stimulation (tDCS), for non-invasive, painless brain stimulation using two electrodes placed over the head which modulates neuronal activity. Although it is still experimental, it has the advantages of being cheap, non-invasive, painless, and safe. Several studies suggest it may be a valuable tool for the treatment of neuropsychiatric conditions such as depression, anxiety, Parkinson's disease, and chronic pain. For the purposes of this section of the paper, research shows cognitive improvement in some patients (Johns Hopkins Hospital n.d.). The results of these treatments in improving cognitive function are important for the functioning and rehabilitation of many patients and are also presumably helpful in improving their capacity to make better informed moral decisions. If results from these therapies continue to improve, then such technologies, provided they are ethically safe and effective in improving cognition, will have a role to play in helping people make better moral decisions.

*4.5. Brain to Computer Implants*

The second technological method that may help people make better moral decisions is through the use of brain-to-computer implants. Although this is presently remote for most people, Kevin Warwick's experiments show its potentially therapeutic use in health care for people with brain injury or mental health problems (Warwick 2014). At the same time, he foresees the possibility of immense strides in cognitive enhancement through this method, aligning with Persson and Savulescu's hope that improving cognitive capacity will also improve human moral awareness and decision-making.

Warwick had a link-to-computer chip implanted in his brain to test his theory. He was then able to control a robot arm in England remotely, from the US, showing implications for future actions and control over machines, presently through extensions of brain-to-computer actions, and possibly in the future, where he thinks brain-to-brain links could be established.

Elon Musk's group, *Neuralink*, continues its work in attempting to merge AI with the human body, claiming we must do so, otherwise we are in danger of becoming inferior to machines that may develop and operate faster than humans. *Neuralink* attempts to link the brain with AI in such a way that AI that would circulate through the veins and arteries using a 'neural lace interface', a wireless brain-computer interface (Hinchcliffe 2018). In other experiments, Charles Lieber of the chemistry department at Harvard says his team is

working to match structural and mechanical properties of electronic and biological systems, and agrees that is should be possible to achieve their integration (Hinchcliffe 2018).

Warwick is enthusiastic about brain-to-computer possibilities in the field of communications: AI is already faster in math, information processing, memory, sensory input, ability to think in more dimensions, etc. These complex machines have a fast-acting networking capacity and we already do not have complete control over them, e.g., we cannot 'switch off' the internet. More importantly for our thesis, he points out that although emotions, feelings, images, and so on, cannot be transmitted from brain-to-brain in their original form; *direct* brain-to-brain communication would enable that. His work, along with others, involves experimenting with radio telegraphic communication between human nervous systems, and this approach looks promising (Warwick 2014).

*4.6. Technology and Improved Moral Thinking*

Concerning the second part of my comments, I question, similarly to Trothen and Buttrey, whether technology might enhance our capacity for better moral thinking and decision-making. That is, will more information and more accurate information be obtainable by, say, computer algorithms that yield more accurate human moral judgments? Savulescu and others hope so, but obviously cannot know this in advance. There are several matters to consider here, in the context of what is usually thought of as moral agency, where the individual exercises personal conscience in the process of decision-making.

*4.7. The Role of Knowledge and Freedom in Moral Agency*

We three authors accept the Christian teaching outlined earlier by Trothen that we are created in God's image. According to Aquinas, being created as such endows us with an innate capacity to know right from wrong, or synderesis (Aquinas, *Summa Theologiae*, 1, Q79), which is the faculty of conscience to resolve situations in a way that respects love for God and love for our neighbour. Although it is highly individual in its innermost workings, Christian conscience formation must look first to objective realities in its assessment of the context of any situation where a decision is to be made.

Contexts often need to be interpreted separately from a subjective viewpoint, but the factual reality of the situation needs to be assessed and clarified (*Aquinas, Summa Theologiae* I-II, Q. 94, a.4). A Christian conscience will take into consideration obedience to God's will, at least as far as we can judge it, as opposed to our own. Conscience is, then, not principally a faculty to promote our own individual progress, but to make an informed judgement about reality. We must look at and assess all the circumstances and the possible short- and long-term consequences of our actions on ourselves and others, as well as deciding whether the means we plan to use are ethical, at the same time taking into account as far as possible our own subjective biases and cultural prejudices, and, most importantly, praying and discerning God's will for us in deciding what is the right thing to do.

4.7.1. Manipulation of the Brain and Moral Agency

In light of this inner work of forming conscience, would we still be truly individual moral agents if a means such as TMS were to be used to an extent that could affect our moral thinking by manipulating the brain? Would the freedom of our agency be compromised, or would the means simply open us up to information to which we would otherwise lack access? If changes through, e.g., TMS, were cognitive only, the latter would apply. Could TMS then help us make better moral decision? Although it has been shown that such methods can improve cognition, it must be remembered that information may not always mean WE will be 'better', since, being human, we will still make mistakes, or despite having cognition improved through therapies, and thus having information pointing towards a good moral decision more available, all human beings sometimes deliberately move in a direction we know to be wrong and against what we know to be right. In Christian language, we would then be 'sinning'. One of the fascinating features of our God-given moral freedom is that we experience the freedom to choose either way, despite perfect (or

improved) knowledge (*Catechism of the Catholic Church* n.1857). It is doubtful that advanced cognition would change that.

A question raised about technological brain stimulation by any of these methods is whether they could be used to subdue traits in a person that *others* find undesirable? What about the 'freedom to fall or fail' argument (Harris 2011)? It could be argued that moral education always involves other people's conclusions, too. That is an inevitable part of education, but another part is that it should help us make our own choices eventually, with the freedom to take a different path from our teachers. The argument about bias in AI is also applicable here and is problematic, since many studies have shown results that are distorted racially, politically, ethnically, regarding gender, and so on, depending on the views of those responsible for the original questions and theses, as well as on methods used for acquiring subjects for the studies. As Trothen notes, morality has a contextual dimension.

### 4.7.2. Moral Agency Composed of Intellect and Will

Theological explorations of virtue, Buttrey claims, are more robust if they include reference to formative historical theological thinkers including Aquinas. Aquinas discusses this linkage of intellect and will in the Summa Theologiae (1, Q 82 and 3). Knowledge of facts and circumstances is needed in the first place in making a moral assessment, but it takes an exercise of the will to execute an action. We know that we can be educated to absorb and appreciate knowledge about right and wrong through logic and understanding, but it is another matter to enact those right or wrong deeds. Aquinas is a good source for discussing the difference between intellectual habits and moral habits, but the 'free will' question continues to challenge philosophers and theologians. Although the debate continues as to whether and when our will is truly free, we experience a sense of completion when we finally execute a deliberate plan of action: 'there, I've done it!' No more time is to be spent wondering when and if I should do what my conscience and self-reflexivity has already formulated for me (the Hamlet dilemma).

### 5. Could Enhancement Enable Us Always to Choose the Moral Good?

Could our capacity for making better moral decisions be enhanced to the extent that we would always choose the moral good? An ongoing challenge here is to know our starting point, as both Trothen and Buttrey indicate. Just as Alasdair MacIntyre asked: *Whose Justice? Which Rationality?* (MacIntyre 1989), we must ask whose and which starting points in ethical thinking should be used? Whose agenda, whose educative tools? Persson and Savulescu are clear that we must work to save the planet before technology produces weapons that are increasingly destructive; therefore, if they were in charge of this moral revolution by biomedical enhancement, those are the outcomes that would be sought through programming.

### 5.1. Varying Views of 'The Moral Good'

A spanner in the works is that experience has shown that, perhaps unfortunately for their agenda, people can and do reach different conclusions, even if they are given the same factual information. Knowledge by itself does not lead to better moral decisions, and society continues to disagree on vital matters such as our responses to the challenges of climate change. We base our decisions on different values, and surely this would happen even if we were morally enhanced by any method?

Persson and Savulescu, however, believe that *more* of us, at least, would think in the same way if we were to be so enhanced, which could affect change. They already have a certain conclusion in mind that would be the point of the procedure: they believe that, with more information about approaching dangers, every one of us will recognize what needs to be done. Perhaps, perhaps not, as noted above. Some people might decide that it would be better if the population were to decrease, through famine or otherwise. Some might see scope for control of resources and personal profit. Some might capitalize on others'

misfortunes. These tendencies have always arisen in times of warfare and both authors acknowledge that a constant human trait persists that finds it easier to harm than to help.

### 5.2. Biotechnological Moral Enhancement and the Replacement of Moral Agency

Russell Hittinger writes: "[b]ut most distinctive of contemporary technology is the replacement of the human act; or, of what the scholastic philosophers called the *actus humanus*. The machine reorganizes and to some extent supplants the world of human action, in the moral sense of the term . . . [There is] a new cultural pattern in which tools are either deliberately designed to replace the human act, or at least have the unintended effect of making the human act unnecessary or subordinate to the machine" (Hittinger 1993).

He adds that, above all else, our culture prizes " . . . the machine insofar as it promises an activity superior to the human act" (Hittinger 1993). Hochschild says, "[i]f that is right, then the threat of automation isn't the bad things it tempts us to do, but its ability to hyp-notize us into thinking we don't even rise to the status of moral agents" (Hochschild 2015). There are certainly fears about the threat of the machine regarding people's employment, being 'taken over', and reduced in our capacities by the superiority of machines in some fields. The playing out of "Human v. Machine' continues to be of great importance for the individual, the common good, and the way we perceive our role in society.

Some claim that Hittenger's traditional interpretation of the actus humanus is reflected in the two hemispheres of the brain, left and right. Although this is not an attempt to produce anything like a full account of this, Hittenger briefly stated that it shows how right brain thinking facilitates appreciation of beauty, life, the sacred, and relationships, whereas left brain thinking demands explicitness, clarity, and rationality (Sharkey 2021). The two spheres are connected and balanced through the neural tissue of the corpus callosum. Psychiatrist/philosopher Iain McGilchrist theorizes that modern people have allowed left brain thinking to predominate, and that this is rampant in many fields of study, especially in the use of artificial intelligence (McGilchrist 2021). AI has the characteristics of a *wunderkind* in dealing with facts and statistics, but it does not possess the balancing effects of right brain characteristics and, therefore, it cannot carry out fully human acts. Pursuit of this type of study is important for developing concepts of human anthropology and spirituality, as well as for considering the question of *human* agency, *human* freedom and *human* wisdom. If machines cannot incorporate these experiences, they will continue to outperform humans in terms of the speed at which they master facts and in patterning, but not in wisdom, spirituality, transcendence, and other experiential realities. A major question lingers: is there any way in which a machine could ever experience the 'numinous?'

### 5.3. Is Biotechnological Moral Enhancement an Imperative?

Some reject this, on the basis that moral traits form the core of a person's identity and that employing technological means of manipulating them could endanger that identity (Huang 2018; Crutchfield 2018). Savulescu and Persson believe use of such a means of manipulation would be to our mutual advantage; therefore, they do see it as an imperative, given present global circumstances. Others think cognitive enhancement could be allowed to complement an already developed virtue of prudence in a person, and that technological means would be permissible to achieve some higher-order desire people wish to attain but cannot reach on their own.

## 6. Concluding Statement

It is clear that further long-term studies are needed on the technological means of achieving either cognitive or moral bio-enhancement, since those currently being un-dertaken remain mostly conjectural until clearer evidence of their effectiveness can be demonstrated.

## 7. A Cautiously Optimistic Recommendation

All three authors agree that better moral thinking and education are good in themselves. Persson and Savulescu tie their desire for enhanced moral thinking to the specific goal of climate change, since they believe time is running out for planet Earth and that we must encourage every means possible, including biomedical and biotechnological enhancement, in an urgent effort to resolve this potential catastrophe.

However, even when faced with this dilemma, ethicists such as ourselves raise additional issues. Trothen cites concerns about the theological meaning of being human, the meaning of choice, and the interdependence of life as this relates to social justice; Buttrey refers to the need for virtue and raises questions concerning elitism and ableism when assessing who will benefit from enhancements; McQueen urges for more attention to be paid to human freedom, moral agency, and the common good, agreeing with the conclusions of her co-authors.

These ethical concerns, along with the necessary risk assessments that must be performed on all pharmaceutical and biotechnologies that aim to bring about cognitive or moral improvement, demand a precautionary approach to their use, whether already in practice, being tested in clinical trials, or in the process of obtaining regulatory approval. A hasty deployment of these enhancements, despite their rapid application urged by people such as Persson and Savulescu, would be unethical until concerns such as those raised here are answered more fully. If these concerns are addressed, BMEs may be theologically justifiable and helpful, *as a supplement*, to moral improvement.

**Author Contributions:** Conceptualization, M.B., M.M. and T.J.T.; methodology, M.B., M.M. and T.J.T.; formal analysis, M.B., M.M. and T.J.T.; investigation, M.B., M.M. and T.J.T.; resources, M.B., M.M. and T.J.T.; writing—original draft preparation, M.B., M.M. and T.J.T.; writing—review and editing, M.B., M.M. and T.J.T. All authors have read and agreed to the published version of the manuscript.

**Funding:** This research received no external funding.

**Institutional Review Board Statement:** Not applicable.

**Informed Consent Statement:** Not applicable.

**Data Availability Statement:** Not applicable.

**Conflicts of Interest:** The authors declare no conflict of interest.

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
