# Peer review of "Faster, Higher, More Moral: Human Enhancement and Christianity"

_religions, doi:10.3390/rel13040354_

Round 1

Reviewer 1 Report

The paper raises a lot of interesting points in is fairly original as far as I know. The paper suffers from too many shortcomings to be published at this point. I raise some points below and other can be found in my comments in the pdf attached. The paper lacks unity. The authors are encouraged to better harmonize their individual work. Some sections are too long (e.g. the extensive discussion on virtues) and others too short (the tension between Christian virtue ethics and BME). The paper could be better embedded in the literature (see suggestions in pdf). Some arguments raised are underdevelopped and therefore unconvincing (see also pdf). The authors also merely present there own opinions as arguments on several occasions.

Author Response

We are grateful to this reviewer for referring us to the article by Hans Van Eyghen (2021). We found this article very helpful. We addressed all of the points identified in the PDF of our mss. We provided some additional references to address the concern regarding opinions versus arguments, and we deleted comments founded mostly on opinion. We also made explicit several connections between the three authors’ sections, thus harmonizing our work and developing better coherence.  Thank you for taking so much time to provide us with this feedback.

Reviewer 2 Report

The article is an interesting paper on an actualy theme of human enhansement in fhe field of moral behaviour from the the perspective of Christian tradition written by three authors. The first author talks about meaning and potential of moral enhancement and cosiders some of the risks and limitations in reference to the human dignity, choice and social justice. The second author takes into consideration the reason of moral education and Thomistic virtues. The third author introduces into the discussion the argument of conscience, freedom and human wisdom. All the parts of this article are discussions with the arguments presented in the book written by philosophers I. Perssson and J. Savaulescu ant their proposal about moral bioenhancement. All of the part of this arcile can be a separate paper on this theme. The reflexions of the first and the third author are well constructed and an interessting discussion on the problem of human enhancement and can be published. The reflections of the second author are not enought logically presented and not enough justified. Thist part of the reviewed paper does not explain clearly the problem and is not well structured like the others parts and stands out of the theme of this article. The reflexions of the second author should be reconsider or omit by the publishing of this paper.

Author Response

Thank you for your constructive comments. As a result, we have made more references to each other’s work to provide more cohesiveness, which we agree was lacking in our discussion of different aspects of moral enhancement.  Also, we have made more explicit connections between pharmaceutical and biotechnological aspects of BME and the ongoing need for training in virtue, even as MBEs become more reliable and accessible. The relevance of author 2’s section is now much clearer.

Reviewer 3 Report

The article deals with interesting and relevant issues but the style of presenting the arguments is somewhat deficient. The article gives the impression that the three authors summarize the findings of other authors and then try to interpret and/or critically reflect on these findings. There is a slight lack of coherence in the paper, as the authors distinguish who writes what and try to put the pieces together. Also, There is a vast array of literature on this subject - I wonder if the authors would consider entering into a critical conversation with more authors.

Author Response

Thank you for your critique pointing out our lack of coherence. We agree that is was lacking and have endeavoured to incorporate more references to each other’s work to make it more integrated. We also agree there is a vast literature on this subject  which we would certainly take into account in future articles. For the purposes of this paper, unfortunately, more references would have made the paper  too long, but hope it will serve as a starting point for people from various faith-filled perspectives.

Reviewer 4 Report

The article addresses the interesting issue of the connection between morality (ethics) and technology. The topic is timely in the context of the development of human technical capabilities. 
The article shows the main lines of the subject discussion.
It can be a starting point for further research and discussion.

Author Response

Thank you for your encouraging comments. We have added emphasis to our objective of stimulating further research and conversation.

Reviewer 5 Report

This article seems to be a classic example of a Catholic approach to innovation,
such as BME, which does not reject innovation a priori, but weighs the objective
pros and cons, assesses compliance with Catholic doctrine, and ultimately takes
an open stance to continue research to reveal additional pros and cons.
By doing so, the article loses some of its originality because it refrains
from the risk of radical novelty, which is sometimes necessary for progress,
and remains within the safe framework of official doctrine.
We see its main value in pointing to the breadth of the Catholic approach to
innovation, as the Church has learned to be cautiously open to the unknown,
including artificial intelligence, on the basis of erroneous moves in the
past when it comes to scientific innovation.
We may have missed the discussion of the problem of revenge (see Jan Assman),
which is the main culprit for people’s bad moral decisions, and for which
Catholicism offers a theoretical and practical solution.

Author Response

Thank you for your insightful review. We are of different religious perspectives (which we have now clarified in the article), but  the Catholic author agrees that the article reflects an open stance to research in this field that follows a general Catholic approach, promoting MBE where possible and calling for clarification on some points through further research. You are right about the risk of the article’s losing some originality in not being more radically novel, but this is not due to Catholic teaching specifically, rather from our own belief about BME, which we view with cautious optimism while indicating areas where further ethical thinking may be required. However, largely as a result of your helpful encouragement regarding the risk of radical novelty, we reached a clearer conclusion advocating for the potential use of moral bioenhancements.

Round 2

Reviewer 1 Report

The paper has improved considerably. I'm still struggling to summarize the overall argument to some extent and the paper is more a call for further reflection than a clear argumentative piece. Despite this, I believe the paper is good enough for publication.